# FedNano: Toward Lightweight Federated Tuning for Pretrained Multimodal Large Language Models

## Abstract

Multimodal Large Language Models (MLLMs) excel in tasks like multimodal reasoning and cross-modal retrieval but face deployment challenges in real-world scenarios due to distributed multimodal data and strict privacy requirements. Federated Learning (FL) offers a solution by enabling collaborative model training without centralizing data. However, realizing FL for MLLMs presents significant challenges, including high computational demands, limited client capacity, substantial communication costs, and heterogeneous client data. Existing FL methods assume client-side deployment of full models, an assumption that breaks down for large-scale MLLMs due to their massive size and communication demands. To address these limitations, we propose **FedNano**, the first FL framework that centralizes the LLM on the server while introducing *NanoEdge*, a lightweight module for client-specific adaptation. *NanoEdge* employs modality-specific encoders, connectors, and trainable *NanoAdapters* with low-rank adaptation. This design eliminates the need to deploy LLM on clients, reducing client-side storage by **95%**, and limiting communication overhead to only **0.01%** of the model parameters. By transmitting only compact *NanoAdapter* updates, *FedNano* handles heterogeneous client data and resource constraints while preserving privacy. Experiments demonstrate that *FedNano* outperforms prior FL baselines, bridging the gap between MLLM scale and FL feasibility, and enabling scalable, decentralized multimodal AI systems.

## 1 Introduction

Multimodal Large Language Models (MLLMs) (Zhu et al., 2023; Liu et al., 2024b; Peng et al., 2023b; Alayrac et al., 2022; Li et al., 2023) excel in tasks like cross-modal retrieval (Yin et al., 2024), making them indispensable for applications such as visual question answering (VQA) (Antol et al., 2015). However, real-world deployment remains fundamentally constrained: multimodal data is inherently decentralized and privacy-sensitive, while the large parameter footprint of MLLMs renders on-device execution infeasible for edge clients.

Federated learning (FL) (McMahan et al., 2017) offers a promising solution for decentralized multimodal training. However, applying FL to MLLMs presents fundamental system-level challenges. First, although parameter-efficient fine-tuning (PEFT) (Houlsby et al., 2019; Lester et al., 2021; Zaken et al., 2021; Hu et al., 2021) reduces the number of trainable parameters, it still requires

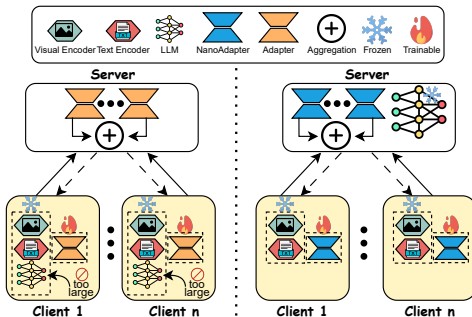

Figure 1: Comparison between traditional PEFT-based FL (left) and our proposed *Fed-Nano* (right). *FedNano* keeps the LLM centralized on the server and performs lightweight tuning on clients, reducing both computation and communication overhead.

deploying the full MLLM—often exceeding 10 billion parameters—on each client, which is impractical for resource-constrained devices such as mobile phones or IoT systems. Second, PEFT

methods typically insert adapters into internal layers of the language model, requiring structural access and full-model execution on clients, as seen in recent FL adaptations such as FedDPA-F (Yang et al., 2024), pFedLoRA (Yi et al., 2023), and FedIT (Zhang et al., 2024). Third, the resulting adapter updates remain sizable, imposing substantial communication overhead across training rounds. Finally, non-IID client data introduces statistical heterogeneity that degrades global model convergence. These limitations collectively constrain the scalability and practicality of existing FL approaches for MLLMs.

To this end, we propose **FedNano**, the first FL framework that enables MLLM adaptation without deploying LLM on clients. As illustrated in Fig. 1, *FedNano* centralizes LLM on the server in a frozen state, and equips each client with *NanoEdge*—a lightweight adaptation module comprising modality-specific encoders, connectors, and trainable *NanoAdapters*. These adapters operate externally to LLM and are optimized using low-rank decomposition (Hu et al., 2021), minimizing both parameter size and transmission cost. This design removes the need for local LLM deployment, reduces

Table 1: Comparison of parameter distribution and communication efficiency between *FedNano* and FedDPA–F (Yang et al., 2024) on LLaVA–1.5–7B (Liu et al., 2024b). *Client Params* denotes parameters retained on client devices, while *Server Uploads* denotes parameter updates sent to the server.

| Approach | Client Params | Server Uploads |
|----------|---------------|----------------|
| *FedNano* | 304.55M (4.30%) | 1.05M (0.01%) |
| FedDPA-F | 7222.81M (100%) | 180.89M (2.50%) |
| **Reduction Rate** | ↓ **95.7%** | ↓ **99.4%** |

client storage by over **95%**, as shown in Tab. 1. Only compact *NanoAdapter* updates are exchanged across training rounds, achieving **over 99% communication reduction** compared to PEFT-based FL methods (Yang et al., 2024). By decoupling adaptation from the LLM, *FedNano* provides a scalable and communication-efficient solution for real-world MLLM deployment.

To address client heterogeneity, *FedNano* adapts Fisher Merging (Matena & Raffel, 2022) to align global updates with client-specific data distributions. This adaptation improves performance on non-IID datasets and outperforms traditional aggregation methods such as FedAvg (McMahan et al., 2017) and FedProx (Li et al., 2020). By integrating these innovations, *FedNano* effectively bridges the gap between the computational complexity of MLLMs and the constraints of FL, enabling efficient deployment in real-world scenarios.

Experiments across diverse MLLM and multimodal tasks demonstrate that *FedNano* not only outperforms existing methods but also significantly reduces resource and communication costs, enabling the scalable, efficient, and privacy-preserving deployment of MLLMs. This framework lays a strong foundation for advancing multimodal AI systems in decentralized real-world applications, including personalized healthcare, cross-device collaboration, and multimodal user interfaces.

The key contributions of this work are:

1. Novel FL Framework for MLLMs: We propose *FedNano*, the first framework that centralizes the LLM on the server and enables lightweight client-side adaptation via *NanoEdge*, reducing client storage by over **95%** and enabling practical deployment on resource-constrained devices.

2. Communication-Efficient Adaptation: *FedNano* employs low-rank decomposition in *NanoAdapters*, achieving an over **99%** reduction in the number of transmitted parameters, allowing efficient deployment in bandwidth-constrained environments.

3. Improved Generalization on Non-IID Data: We adapt Fisher Merging for FL, aligning global updates with client-specific distributions to improve model performance on heterogeneous datasets.

4. Comprehensive Validation: Extensive experiments demonstrate the effectiveness and efficiency of *FedNano*, establishing it as a scalable solution for real-world MLLM deployment.

## 2 RELATED WORK

### 2.1 MULTIMODAL LARGE LANGUAGE MODELS

MLLMs (Zhu et al., 2023; Liu et al., 2024b; Peng et al., 2023b; Alayrac et al., 2022; Li et al., 2023; Dai et al., 2023) extend LLMs (Touvron et al., 2023; Peng et al., 2023a; Bai et al., 2023) by

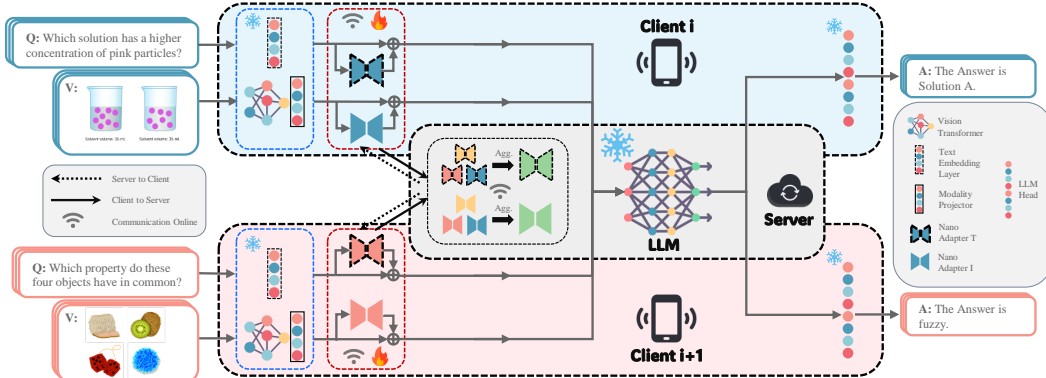

Figure 2: Overview of the *FedNano* framework. The server hosts the frozen LLM, while each client performs local tuning via *NanoEdge*, which includes *NanoAdapter*-T for text and *NanoAdapter*-I for vision. Clients upload low-rank adapter updates, which are aggregated on the server using Fisher merging. This design reduces client overhead and supports scalable, multimodal federated learning under data heterogeneity.

integrating modality-specific encoders and connectors to process multimodal inputs. Recent works focus on efficient alignment, using lightweight connectors such as the linear projection in MiniGPT-4 (Zhu et al., 2023) or the MLP bridge in LLaVA (Liu et al., 2024b). However, these models assume full model access, which is incompatible with federated settings due to privacy and resource constraints. *FedNano* resolves this by freezing the LLM on the server and enabling lightweight client-side adaptation via *NanoAdapters*.

### 2.2 PARAMETER EFFICIENT FINE-TUNING

PEFT techniques (Houlsby et al., 2019; Lester et al., 2021; Zaken et al., 2021; Hu et al., 2021) adapt large pretrained models by updating only a small set of parameters, significantly reducing training costs. They include additive methods like adapters (Houlsby et al., 2019) and soft prompts (Lester et al., 2021), selective tuning such as BitFit (Zaken et al., 2021), and reparameterization methods like LoRA (Hu et al., 2021). While effective in centralized settings, PEFT-based FL methods (Chen et al., 2023; Wang et al., 2024; Zhang et al., 2024; Bai et al., 2024; Liu et al., 2025; Hu et al., 2025) assume the full model, including LLM, can be deployed on clients. This becomes impractical for MLLMs, where LLM accounts for the vast majority of parameters and cannot be hosted on resource-limited devices. To overcome this, *FedNano* introduces a new paradigm: the LLM is frozen and centralized on the server, while lightweight *NanoAdapters* are deployed on clients. This design eliminates the need for full-model access, reduces client overhead, and enables scalable FL for MLLM. Unlike conventional PEFT, which are inserted into LLM, *NanoAdapters* operate externally, interfacing solely through the modality connector. This allows adaptation without modifying or executing LLM on clients.

### 2.3 MULTIMODAL FEDERATED LEARNING

Multimodal FL has gained increasing attention for handling data heterogeneity and privacy constraints in real-world deployments. Prior work has focused on vision-language models, proposing strategies for modality imbalance (Yu et al., 2023; Che et al., 2024), non-IID distributions (Yang et al., 2024; Zhang et al., 2024; Chen & Zhang, 2024), and client personalization (Yi et al., 2023; Chen et al., 2023). Benchmarks like FedMultimodal (Feng et al., 2023) and FedMLLM (Xu et al., 2024) further standardize evaluation in heterogeneous multimodal settings. However, these methods still rely on client-side full model deployment. For MLLMs, this becomes infeasible due to their scale. Even with PEFT, deploying full MLLMs locally remains out of reach, and transmitting adapter updates still incurs significant communication overhead. *FedNano* departs from this design by keeping the LLM on the server and transmitting only compact *NanoAdapter* updates from clients. This makes it the first scalable FL framework tailored for large-scale MLLMs, enabling efficient multimodal collaboration without sacrificing practicality.

## 3 METHODOLOGY

### 3.1 PROBLEM DEFINITION

This work addresses federated fine-tuning for MLLMs in decentralized environments with statistical data heterogeneity. Each client $k$ holds a private multimodal dataset $D_k = \{(v_k^i, q_k^i, a_k^i)\}$, comprising image-question-answer triplets. We assume complete modality availability and a shared model architecture across all clients; only data distributions differ (Chen et al., 2023). The marginal distributions of $v_k^i$, $q_k^i$, and $a_k^i$ vary across clients, resulting in shifts in both visual and textual representations, as well as answer semantics. Such heterogeneity poses challenges for achieving consistent generalization, as standard aggregation strategies struggle to align diverse local updates.

Our objective is to collaboratively fine-tune a shared global MLLM for VQA (Antol et al., 2015). Following (Liu et al., 2024a), we formulate this as an open-ended generation problem, where the model generates free-form answers given image-question pairs. Existing approaches assume that the full MLLM can be deployed on each client, which is infeasible in practice due to the massive size of LLM backbones. Client devices often lack sufficient compute, memory, and bandwidth to support such models, and privacy regulations further restrict centralized data access. These constraints call for a new FL framework that avoids client-side LLM deployment while enabling efficient adaptation and communication. To address these challenges, we propose **FedNano**, a parameter-efficient framework that centralizes the computationally intensive LLM on the server while enabling lightweight, client-specific tuning. In the following sections, we detail the design of *FedNano*, focusing on how it minimizes computational and communication overhead and addresses data heterogeneity.

### 3.2 OVERVIEW OF *FedNano* ARCHITECTURE

*FedNano* is designed to address the key challenges of deploying MLLMs in FL environments. As shown in Fig. 2, it introduces a new architecture that centralizes the computationally intensive LLM on the server, while clients retain only lightweight *NanoEdge* modules for task-specific adaptation. *NanoEdge* freezes the modality encoders and connector, and trains only the *NanoAdapters*, which are small, efficient modules inserted at the interface with the LLM. Each modality is equipped with its own adapter, enabling modular and decoupled adaptation across vision, text, or other modalities. This design eliminates the need to deploy the full model on resource-constrained devices, significantly reducing client-side computation and enabling edge deployment on mobile or IoT systems. It also allows clients with missing modalities to participate in training by updating only the available adapters, and supports seamless extension to new modalities by simply plugging in additional adapters without modifying the existing architecture. The complete training and aggregation process is detailed in the Appendix.

*FedNano* jointly addresses three key challenges in MLLM-based FL: high computation, communication cost, and data heterogeneity. By offloading the LLM to the server, clients train only the *NanoEdge* module, which includes frozen encoders and a connector, and optimizes a small set of *NanoAdapters* for task-specific adaptation. The total client-side module accounts for less than **5%** of the model parameters, while the trainable *NanoAdapters* comprise only **0.01%**. During aggregation, only *NanoAdapters* updates are uploaded, significantly reducing communication overhead. *NanoAdapters* are optimized via low-rank decomposition, enabling expressive local tuning while preserving pretrained alignment with the frozen LLM. This compact update mechanism supports low-bandwidth environments and enhances training efficiency. To address data heterogeneity, *FedNano* integrates Fisher Merging (Matena & Raffel, 2022) into FL as an advanced aggregation strategy, leveraging client-specific posterior estimates to align local updates with global objectives. By weighting and combining *NanoAdapter* updates based on their estimated importance, this method improves robustness across diverse datasets, even under non-IID conditions. Together with its architectural and optimization designs, *FedNano* bridges the gap between the computational barriers of MLLM deployment and the practical constraints of FL, offering a scalable, efficient, and privacy-preserving solution for decentralized multimodal learning.

### 3.3 *NanoEdge*: CLIENT-SIDE TUNING MODULE

MLLMs are composed of three key components: modality encoders, a connector, and a pretrained LLM backbone. The modality encoders extract embeddings from raw inputs, such as images and

text, while the connector aligns these embeddings into a unified representation compatible with the LLM. Together, these components enable MLLMs to effectively handle diverse multimodal tasks by leveraging their pretrained capabilities.

Building on this structure, *NanoEdge* introduces *NanoAdapters* at the interface between the connector and the LLM to facilitate efficient task-specific tuning while preserving the pretrained alignment across modalities. By freezing the modality encoders and the connector, *NanoEdge* maintains their alignment with the LLM, ensuring the foundational structure of the pretrained model remains intact. This design allows *NanoAdapters* to focus solely on learning task-specific patterns from local client data and integrating federated knowledge updates, avoiding any disruption to the pretrained alignment. By restricting training to the lightweight *NanoAdapter* parameters, *NanoEdge* minimizes client-side computational demands while enabling efficient and privacy-preserving adaptation.

The *NanoAdapters* employ a low-rank decomposition mechanism, inspired by LoRA (Hu et al., 2021), consisting of a down-projection to reduce embedding dimensionality and an up-projection to restore it. This design balances parameter efficiency and adaptation capability, enabling *NanoEdge* to perform localized tuning and transmit updates efficiently. Each modality is equipped with a dedicated *NanoAdapter*—$\mathcal{A}_I$ for images and $\mathcal{A}_T$ for text—capturing modality-specific patterns essential for multimodal tasks. Unlike traditional adapters that are inserted into LLM, *NanoAdapters* remain externally attached to the modality connector, requiring no structural access to or execution of LLM. This makes them uniquely compatible with server-hosted LLMs in federated environments.

### 3.4 Fisher-Guided Adaptive Aggregation

In FL, model aggregation can be interpreted as maximizing the joint posterior likelihood across clients. Traditional methods like FedAvg implicitly assume isotropic Gaussian posteriors (Matena & Raffel, 2022), which oversimplifies client uncertainty and leads to degraded performance under data heterogeneity. *FedNano* addresses this limitation by adopting Fisher Merging (Matena & Raffel, 2022), which leverages the Laplace approximation for more accurate posterior estimation. The global update is computed as:

$$\theta_{global} = \frac{\sum_{k=1}^{K} \frac{|D_k|}{\sum_{k=1}^{K} |D_k|} F_k \theta_k}{\sum_{k=1}^{K} \frac{|D_k|}{\sum_{k=1}^{K} |D_k|} F_k}, \tag{1}$$

where $\theta_k$ denotes the *NanoAdapter* parameters of client $k$, $F_k$ is the Fisher Information Matrix (FIM), which serves as the precision matrix of the Laplace approximation, and $D_k$ is the local dataset. This weighting improves the alignment of local updates with their estimated importance, enhancing generalization under non-IID data. To ensure scalability, *FedNano* approximates the full FIM with its diagonal (Kirkpatrick et al., 2017), and computes it efficiently from squared gradients during backpropagation (Wu et al., 2023), reducing computation from $O(|\theta|^2)$ to $O(|\theta|)$ without sacrificing aggregation accuracy. Moreover, the FIM is computed independently for each *NanoAdapter*, enabling modality-specific aggregation. This design allows the system to adapt to the distinct statistical characteristics and heterogeneity levels across different modalities. Compared to uniform averaging, this method dynamically prioritizes impactful updates, achieving stronger global performance under statistical heterogeneity.

## 4 Experiment

### 4.1 Experimental Setup

We evaluate our approach on the Visual Question Answering (VQA) task using two established benchmarks: ScienceQA (Lu et al., 2022) and IconQA (Lu et al., 2021). These datasets were selected for their well-defined categorical structures and multimodal complexities, making them particularly suitable for assessing the performance of FL in non-IID settings. To simulate FL in a non-IID setting, we partitioned the datasets using Dirichlet distributions following (Che et al., 2023; Lai et al., 2022; Zhang et al., 2024) with a concentration parameter $\alpha = 1$ to create strongly non-IID splits. Partitioning was guided by topic annotations in ScienceQA and skill annotations in IconQA, ensuring heterogeneous yet meaningful distributions across five simulated clients. Each partition, representing an individual client dataset, maintains consistent train-validation-test splits

Table 2: Performance comparison of centralized training, local fine-tuning, and federated approaches in ScienceQA and IconQA. *FedNano* consistently achieves top average performance across most clients and both datasets, demonstrating its effectiveness in handling client heterogeneity.

| Backbone | Approach | ScienceQA (Clients) | | | | | | IconQA (Clients) | | | | | |
|---|---|---|---|---|---|---|---|---|---|---|---|---|---|
| | | C1 | C2 | C3 | C4 | C5 | Avg | C1 | C2 | C3 | C4 | C5 | Avg |
| MiniGPT-4 | Centralized | 73.70 | 88.34 | 89.83 | 84.52 | 87.41 | 84.76 | 80.76 | 86.62 | 81.16 | 82.74 | 85.36 | 83.33 |
| | LocFT | 67.74 | 74.69 | 77.42 | 72.46 | 74.07 | 73.28 | 67.70 | 73.48 | 70.63 | 70.86 | 77.53 | 72.04 |
| | FedAvg | 70.22 | 79.65 | 79.65 | 75.19 | 75.56 | 76.05 | 70.31 | 75.61 | 74.98 | 72.76 | 81.25 | 74.98 |
| | FedProx | 70.97 | 80.40 | 80.15 | 75.19 | 75.80 | 76.50 | 70.94 | 77.36 | 74.58 | 71.50 | 80.70 | 75.01 |
| | FedDPA-F | **71.96** | 78.41 | **81.14** | 76.42 | 75.80 | 76.75 | 70.94 | **77.91** | 74.51 | 73.08 | 80.30 | 75.35 |
| | *FedNano* | 68.98 | **81.89** | 80.89 | **76.43** | **77.04** | **77.05** | **72.21** | 77.28 | **75.85** | **74.27** | **82.52** | **76.42** |
| LLaVA-1.5 | Centralized | 83.87 | 91.07 | 89.33 | 90.57 | 89.38 | 88.84 | 86.62 | 88.92 | 84.88 | 87.25 | 88.45 | 87.22 |
| | LocFT | 71.96 | 80.89 | 76.92 | 79.65 | 75.80 | 77.04 | **75.93** | 78.94 | 72.53 | 74.35 | 76.50 | 75.65 |
| | FedAvg | 73.20 | **84.37** | 83.62 | 82.13 | **80.49** | 80.76 | 71.18 | 79.89 | 76.80 | **77.51** | **83.23** | 77.72 |
| | FedProx | 73.95 | **84.37** | 83.87 | 81.39 | 80.00 | 80.71 | 70.23 | 80.13 | 76.72 | **77.51** | 82.36 | 77.39 |
| | FedDPA-F | 73.70 | 84.12 | 84.12 | 81.89 | 79.51 | 80.67 | 72.12 | 79.65 | 76.80 | 77.43 | 82.36 | 77.68 |
| | *FedNano* | **74.94** | 84.12 | **84.86** | **82.88** | 80.25 | **81.41** | 72.13 | **80.44** | **77.36** | 77.43 | 82.83 | **78.04** |

Table 3: Performance of MiniGPT-4 on IconQA with 10 simulated clients. *FedNano* achieves the highest average accuracy across all clients, demonstrating strong scalability and consistent effectiveness as the federated environment becomes more fragmented, reinforcing its practicality for large-scale real-world deployments.

| Approach | C1 | C2 | C3 | C4 | C5 | C6 | C7 | C8 | C9 | C10 | Avg |
|---|---|---|---|---|---|---|---|---|---|---|---|
| LocFT | 67.56 | 69.77 | 73.89 | 67.24 | 79.90 | 72.15 | 69.77 | 64.71 | 71.67 | 67.35 | 70.40 |
| FedAvg | 74.52 | 81.01 | 78.00 | 78.63 | 85.91 | 79.90 | 75.94 | 75.63 | 70.90 | 77.86 | 77.83 |
| FedProx | 73.89 | 76.74 | 77.37 | 75.63 | 84.01 | 76.58 | 73.41 | 71.36 | 78.79 | 72.29 | 76.00 |
| FedDPA-F | 74.52 | 81.01 | 78.00 | 78.63 | 85.91 | 79.90 | 75.94 | 75.63 | 70.90 | 77.86 | 77.83 |
| *FedNano* | **77.03** | **82.77** | **78.22** | **79.67** | **88.57** | **80.35** | **81.34** | 72.84 | **73.77** | **79.47** | **78.86** |

for evaluation. We evaluate our approach on MiniGPT-4 (Zhu et al., 2023) and LLaVA-1.5 (Liu et al., 2024b).

## 4.2 IMPLEMENTATION DETAILS

**Baselines.** To the best of our knowledge, *FedNano* is the first FL framework specifically designed to support MLLMs by centralizing the LLM on the server. This architectural shift renders existing PEFT-based FL methods inapplicable, as they assume full-model access and local integration with the LLM. Given the absence of prior work addressing this setting, we evaluate *FedNano* against three representative FL baselines: FedAvg (McMahan et al., 2017), a foundational aggregation method with limited handling of data heterogeneity; FedProx (Li et al., 2020), which mitigates client drift through a proximal term but lacks parameter-specific adaptation; and FedDPA-F (Yang et al., 2024), which integrates advanced alignment strategies but incurs high computational and communication overheads. We further include comparisons with a centralized model, representing the performance upper bound achieved with access to all data, and locally fine-tuned models, which operate in isolation without collaboration. All results are averaged over 3 runs.

**Training Configurations.** The training process includes 10 communication rounds ($R = 10$), with each client performing one local epoch per round using a batch size of 8. All experiments were conducted on NVIDIA A100 80G GPUs.

Table 4: Performance of MiniGPT-4 on IconQA under different data heterogeneity levels. *FedNano* consistently achieves the highest average accuracy, especially under severe non-IID settings, i.e., $\alpha = 0.1$, highlighting the effectiveness of its Fisher-guided aggregation in aligning heterogeneous client updates.

| Approach | $\alpha = 0.1$ | | | | | | $\alpha = 5$ | | | | | |
|---|---|---|---|---|---|---|---|---|---|---|---|---|
| | C1 | C2 | C3 | C4 | C5 | Avg | C1 | C2 | C3 | C4 | C5 | Avg |
| LocFT | 69.94 | 75.80 | 75.48 | 73.18 | 77.00 | 74.28 | 65.71 | 70.62 | 71.41 | 72.76 | 70.64 | 70.22 |
| FedAvg | 72.80 | 76.80 | 75.50 | 73.20 | 73.60 | 74.38 | 74.34 | 75.61 | 72.92 | **76.08** | 74.68 | 74.72 |
| FedProx | 71.54 | 74.79 | 74.15 | 69.72 | 70.06 | 73.05 | 68.48 | 70.30 | 70.15 | 70.15 | 71.04 | 70.02 |
| FedDPA-F | 70.25 | 76.40 | 74.10 | 72.50 | **78.55** | 74.27 | 71.52 | **76.83** | **74.51** | 73.24 | **75.84** | 74.38 |
| *FedNano* | **73.85** | **78.22** | **80.14** | **76.28** | 74.94 | **76.68** | **74.90** | 76.16 | 74.18 | 74.82 | 73.73 | **74.75** |

## 4.3 MAIN RESULTS

Results in Tab. 2 demonstrate that FL methods consistently outperform locally fine-tuned models (LocFT), emphasizing the benefit of global knowledge sharing in distributed, heterogeneous settings.

*FedNano* achieves the highest average performance among all FL methods, more effectively narrowing the gap to centralized training than existing baselines. While FedAvg performs competitively with simple weighted averaging, its inability to adapt to non-IID data results in suboptimal performance under heterogeneous distributions. FedProx mitigates client drift by constraining local updates toward the global model, but this rigid constraint limits flexibility, making it insufficient for complex multimodal tasks. FedDPA-F, though designed for personalization, requires careful tuning of global training epochs and risks overwriting the global adapter during local updates, potentially degrading performance due to catastrophic forgetting.

In contrast, the superior performance of *FedNano* is attributed to its novel design and optimization strategies. As shown in Tab. 2, *FedNano* achieves an average accuracy of 77.05% on ScienceQA and 76.42% on IconQA for MiniGPT-4, exceeding FedAvg and FedProx, indicating improved generalization in heterogeneous client environments. For LLaVA, *FedNano* attains 81.41% on ScienceQA and 78.04% on IconQA, surpassing FedDPA-F and FedProx, demonstrating enhanced robustness in multimodal FL. These results validate the effectiveness of *NanoAdapters* for modality-specific adaptation, while substantially reducing client-side computational and storage demands, enabling deployment on resource-limited devices. Moreover, *FedNano* integrates Fisher Merging with a diagonal approximation of the FIM, allowing the system to prioritize critical parameter updates based on client-specific confidence. This results in more effective aggregation than uniform averaging, improving stability under non-IID distributions while reducing overfitting to local client noise. By balancing generalization and personalization, *FedNano* consistently delivers strong performance across diverse client settings, all while maintaining minimal communication overhead.

## 4.4 ANALYSIS

**Scalability to Larger Client Populations.** To evaluate the scalability of *FedNano*, we extend the number of clients from 5 to 10 on the IconQA dataset using the MiniGPT-4 backbone. As shown in Tab. 3, *FedNano* achieves the highest average accuracy, consistently outperforming all baselines. This demonstrates that the framework retains its effectiveness even as the federated environment becomes more fragmented. The results confirm that *FedNano* scales robustly with increasing client population, reinforcing its practicality for real-world large-scale federated deployments.

**Robustness under Data Heterogeneity.** To assess the robustness of *FedNano* under varying levels of data heterogeneity, we evaluate its performance on IconQA using the MiniGPT-4 backbone across different Dirichlet concentration values ($\alpha = 0.1$ and $\alpha = 5$). As shown in Tab. 4, *FedNano* consistently achieves the highest average accuracy in the highly non-IID setting ($\alpha = 0.1$), outperforming all FL baselines. This demonstrates the effectiveness of its Fisher-guided aggregation in aligning heterogeneous client updates. While the performance gap narrows under near-IID condi-

tions ($\alpha = 5$), *FedNano* remains competitive, indicating that its advantages are most pronounced in realistic heterogeneous federated scenarios.

**Generalization under Cross-Task Client Distribution.** We evaluate *FedNano* in a challenging cross-task setup where four clients are respectively assigned A-OKVQA, OK-VQA, IconQA, and GQA, introducing significant task-level heterogeneity. As shown in Tab. 5, *FedNano* achieves stable and strong performance across all clients. This robustness stems from its modular design and Fisher-guided aggregation, which enable effective alignment of heterogeneous updates and support generalization across semantically diverse tasks.

Table 5: Performance of MiniGPT-4 in a cross-task federated setup with clients assigned to different VQA benchmarks (A-OKVQA, OK-VQA, IconQA, GQA). *FedNano* achieves the highest average accuracy, demonstrating strong generalization across semantically diverse tasks.

| Approach | C1 | C2 | C3 | C4 | Avg |
|---|---|---|---|---|---|
| FedAvg | 34.35 | 28.83 | 29.00 | 29.53 | 30.86 |
| FedProx | 52.45 | 50.82 | 59.80 | 42.15 | 51.30 |
| FedDPA-F | 52.76 | 51.12 | 60.10 | 42.46 | 51.61 |
| *FedNano* | **54.20** | **52.60** | **60.36** | **43.32** | **52.62** |

**The Necessity of Combining Both $\mathcal{A}_T$ and $\mathcal{A}_I$.** To evaluate the necessity of the textual adapter $\mathcal{A}_T$ and the visual adapter $\mathcal{A}_I$, we conduct ablation experiments using three configurations: $\mathcal{A}_T$ only, $\mathcal{A}_I$ only, and both. For MiniGPT-4, $\mathcal{A}_T$ achieves 45.91% on ScienceQA and 57.77% on IconQA, while $\mathcal{A}_I$ improves to 74.57% and 75.17%. Their combination further boosts accuracy to 76.42% and 76.04%, outperforming $\mathcal{A}_I$ alone by +1.85% and +0.87%. As shown in Tab. 6, similar trends are observed with LLaVA-1.5, confirming the robustness of combining both adapters. The poor performance of $\mathcal{A}_T$ alone suggests that textual inputs

Table 6: Ablation results of different adapter types. Combining $\mathcal{A}_T$ and $\mathcal{A}_I$ yields consistently superior performance, indicating their complementary effects across both backbones.

| Backbone | Variant | ScienceQA | IconQA |
|---|---|---|---|
| MiniGPT-4 | $\mathcal{A}_T$ | 45.91 | 57.77 |
| | $\mathcal{A}_I$ | 74.57 | 75.17 |
| | $\mathcal{A}_T + \mathcal{A}_I$ | **76.42** | **76.04** |
| LLaVA-1.5 | $\mathcal{A}_T$ | 50.08 | 48.15 |
| | $\mathcal{A}_I$ | 77.03 | 77.12 |
| | $\mathcal{A}_T + \mathcal{A}_I$ | **78.04** | **77.83** |

provide insufficient task-relevant information in these vision-centric VQA tasks. These results validate the dual-adapter design of *NanoEdge*, where $\mathcal{A}_I$ handles visual adaptation and $\mathcal{A}_T$ enhances generalization.

**Frequent Communication Amplifies the Advantages of *FedNano*.** As shown in Fig. 3, reduced communication frequency leads to a general decline in global model performance across all methods due to increased parameter divergence, which hinders effective aggregation. Importantly, the results highlight that *FedNano* outperforms FedAvg by a larger margin when communication is more frequent. With shorter intervals, FIM mechanism of *FedNano* can better leverage aligned client parameters to prioritize impactful updates, amplifying its advantages in handling data heterogeneity. In contrast, FedAvg struggles with parameter divergence regardless of communication frequency, showing minimal improvement with more frequent updates. These findings underscore that while frequent communication benefits all methods, it significantly enhances the effectiveness of *FedNano*, reinforcing its superior ability to integrate client-specific updates and maintain robust performance in federated learning environments.

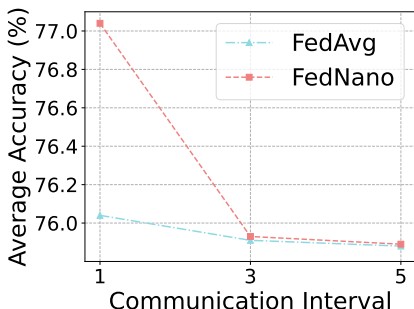

Figure 3: Impact of communication frequency (measured in local epochs per communication round). *FedNano* outperforms FedAvg, with more frequent communication amplifying its advantages.

**Higher Adapter Ranks Enhance *FedNano* Performance.** Fig. 4 illustrates the impact of adapter rank, comparing *FedNano* with FedAvg on the ScienceQA dataset. As the adapter rank increases, accuracy improves due to the enhanced capacity to encode task-specific and client-specific information, which is particularly important in non-IID settings. However, higher ranks also incur greater communication costs, necessitating a trade-off between performance and resource efficiency in FL. *FedNano* consistently outperforms FedAvg across all ranks, with the performance gap widening at higher ranks. This improvement is driven by the FIM aggregation, which leverages richer client-specific updates at higher ranks to achieve better alignment between local contributions and the global model. In contrast, at lower ranks, the limited adapter capacity constrains the quality of updates, reducing the effectiveness of FIM aggregation.

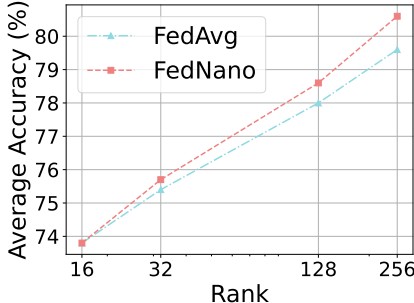

Figure 4: Effect of adapter rank. *FedNano* consistently achieves superior performance, demonstrating its ability to capture task-specific and client-specific information effectively.

## 5 CONCLUSION

This work introduced *FedNano*, an FL framework that tackles the unique challenges of deploying MLLMs in decentralized settings. By centralizing the LLM on the server and employing lightweight *NanoAdapters* on clients, *FedNano* achieves significant gains in both resource and communication efficiency, while effectively addressing data heterogeneity in non-IID environments. Comprehensive evaluations on ScienceQA and IconQA benchmarks demonstrate that *FedNano* consistently outperforms state-of-the-art FL baselines, further narrowing the gap between federated and centralized training. By combining scalable design with robust performance, *FedNano* offers a practical and privacy-preserving solution, advancing the real-world deployment of MLLMs.

## 6 LIMITATION AND FUTURE WORK

While *FedNano* demonstrates strong performance and efficiency, several aspects remain open for improvement. One limitation is the assumption that all clients have similar hardware capabilities to manage *NanoAdapters*, which may not hold in practice. Future work could explore adaptive mechanisms that adjust adapter configurations based on client-specific constraints. Real-world deployments may also involve incomplete modality settings, where some clients lack certain inputs. Thanks to its modular design, each *NanoAdapter* operates independently of the LLM and other modalities, allowing clients to update only the available components. Fisher-guided aggregation on the server side is modality-agnostic, supporting flexible integration of asymmetric updates. These features make *FedNano* naturally extensible to missing or partially labeled modalities, as well as new combinations such as audio-text, audio-visual, or sensor-language, without modifying the core framework. Finally, while *FedNano* already ensures strong privacy through lightweight updates, incorporating techniques like differential privacy could further enhance its guarantees, provided efficiency is maintained.

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
