# OpenReview forum: "FedNano: Toward Lightweight Federated Tuning for Pretrained Multimodal Large Language Models"
_ICLR.cc/2026/Conference — Submitted to ICLR 2026_

### Official Review · Reviewer_B9BP · 2025-10-30

**Soundness:** 2
**Presentation:** 2
**Contribution:** 2
**Rating:** 2
**Confidence:** 5

**Summary:**

The authors propose FedNano, a framework designed to address the challenge of training multimodal large language models (MLLMs) under data privacy constraints within a federated learning (FL) setting. In this approach, the large language model (LLM) is centralized on the server, while each client independently trains lightweight LoRA modules for its own modalities. This design effectively alleviates communication overhead and the limited computational capacity of clients. During server-side aggregation, FedNano employs a Fisher Merging strategy to efficiently integrate LoRA updates from different clients with the central LLM. Experimental results on visual question answering (VQA) tasks demonstrate that this method achieves effective federated multimodal learning while preserving data privacy.

**Strengths:**

The combination of MLLMs and FL is a topic of practical value.

**Weaknesses:**

1. The main innovation claimed in the paper has already been explored in reference [a]. The authors state in the abstract that their work is “the first FL framework that centralizes the LLM on the server while introducing NanoEdge, a lightweight module for client-specific adaptation,” but this claim is inaccurate. Reference [a] adopts a similar approach, centralizing the LLM on the server and even pretraining it with public data to fully leverage server-side computational resources.

[a] MLLM-LLaVA-FL: Multimodal Large Language Model Assisted Federated Learning, WACV, 2025.

2. There are already many lightweight LLMs that can be deployed directly on mobile devices, so the authors’ assertion that the LLM must reside exclusively on the server is not entirely justified.

3. The proposed method also has conceptual limitations. If the LoRA-like adapters on clients are trained solely on local data without aligning or integrating with the LLM weights, and are only merged later through Fisher merging, the performance is likely to be suboptimal. The limited improvement observed in the experimental results compared with other methods seems to support this concern.

4. Although the authors emphasize that their method mainly reduces communication and computation costs, they do not provide quantitative experiments or evidence to substantiate these claims, which should be an essential part of the evaluation.

**Questions:**

The main issue is that the authors’ claim of being the first to deploy the LLM solely on the server is incorrect, as prior work [a] has already adopted this approach and trained only the other modality encoders, which is more reasonable than training separate adapters for the LLM. Moreover, the experimental performance is mediocre, and there is no comparison or discussion of computational and communication costs.

---

### Official Review · Reviewer_2c4M · 2025-10-30

**Soundness:** 4
**Presentation:** 4
**Contribution:** 2
**Rating:** 6
**Confidence:** 4

**Summary:**

FedNano proposes a practical way to fine-tune multimodal LLMs in federated settings by keeping the large LLM frozen on the server and pushing only a lightweight client module, “NanoEdge,” which uses modality-specific encoders/connectors plus low-rank NanoAdapters for tuning. This design removes the need to deploy the full LLM on devices, cutting client storage by ~96% and slashing communication to ~0.01% of model parameters (≈99% fewer transmitted parameters than PEFT-style FL). To cope with non-IID client data, FedNano aggregates updates with Fisher-guided merging (diagonal FIM), improving robustness versus FedAvg/FedProx/FedDPA-F. On ScienceQA and IconQA (with MiniGPT-4 and LLaVA-1.5 backbones), it achieves the best average FL performance, scales to more clients, and generalizes across different VQA tasks—all while remaining communication- and compute-efficient.

**Strengths:**

1. A new FL architecture for MLLMs that keeps the LLM frozen on the server and lets clients adapt via a lightweight “NanoEdge,” cutting client storage by >95%.
2. Communication-efficient adaptation using low-rank NanoAdapters, reducing transmitted parameters > 99%.

**Weaknesses:**

1. The experiments results looks weird to the reviewer, why it looks random as you change the number of clients.
2. The paper is basically combining several techniques which leads a lack of novelty. But the final pipeline is practical so this does not look like a big issue to the reviewer.

**Questions:**

See weaknesses.

---

### Official Review · Reviewer_GqCF · 2025-10-31

**Soundness:** 3
**Presentation:** 3
**Contribution:** 3
**Rating:** 4
**Confidence:** 5

**Summary:**

This paper proposes FedNano, a federated learning (FL) framework for multimodal large language models (MLLMs). Instead of deploying full LLMs on clients, FedNano centralizes the LLM on the server and introduces NanoEdge, a lightweight client module composed of modality-specific encoders and LoRA-based adapters.

**Strengths:**

The paper tackles an important and timely problem—enabling scalable FL for large multimodal models under resource constraints.

**Weaknesses:**

1.While the paper addresses an important problem, its novelty claim is somewhat overstated. The idea of centralizing the LLM on the server while training lightweight client modules is not new—for instance, MLLM-LLaVA-FL[ref1] already adopts a similar design. The paper should clarify how its approach differs from such prior work.

2. Since the client-side adapters are trained locally without interacting with or being guided by the LLM’s frozen weights, it is unclear whether the learned representations remain aligned with the global model semantics. This weak coupling makes the subsequent Fisher-based merging on the server seem heuristic rather than principled.

3. The improvements in model performance are modest, particularly in Table 2.

[ref1] MLLM-LLaVA-FL: Multimodal Large Language Model Assisted Federated Learning. WACV 2025.

**Questions:**

1.Why is Fisher information an appropriate choice for merging NanoAdapters across clients? Are there empirical comparisons with other merging strategies?

2.Have you analyzed failure cases, or tasks where FedNano underperforms compared to full-model FL?

---

### Official Review · Reviewer_jTaq · 2025-11-07

**Soundness:** 3
**Presentation:** 3
**Contribution:** 2
**Rating:** 4
**Confidence:** 4

**Summary:**

This paper tackles the challenge of federated learning for large multimodal models (MLLMs), where deploying full LLMs on clients is infeasible due to computational and communication constraints. The authors propose FedNano, a FL framework that centralizes the LLM on the server (frozen) and deploys lightweight, modality-specific NanoAdapters on clients. Each client updates only these low-rank adapters, while server-side aggregation is guided by Fisher Information-based merging, addressing data heterogeneity.

**Strengths:**

1.	This paper proposes the first FL paradigm that avoids placing LLMs on clients while enabling collaborative tuning for MLLMs, which is both practically motivated and technically interesting.
2.	This work quantitatively reduces client parameters and communication volume by 95%+ and 99%+, respectively. The improvements are clearly reported.
3.	Despite limited trainable parameters, FedNano achieves higher or comparable accuracy to heavier baselines across datasets, backbones, and client numbers.

**Weaknesses:**

1.	Although this work has a clear modular architecture (LLM on server + NanoEdge on clients). Key design choices such as the exact placement of NanoAdapters, adapter architecture (rank, dimension per modality), and training pipeline (forward/backward interaction with server LLM) remain insufficiently described for full reproducibility.
2.	The Fisher-guided merging is claimed as a core innovation, however, there is no explicit ablation of “FedNano w/o Fisher” vs “with Fisher”, making it unclear how much performance gain comes from NanoAdapter vs aggregation strategy.
3.	Despite the significant 99.4% reduction in communicated parameters, FedNano still outperforms FedDPA-F in accuracy, which is counterintuitive. Although the authors provide some empirical explanations in the experimental analysis, these justifications lack theoretical grounding and solid evidence, making the conclusion insufficiently convincing.
4.	Experiments focus only on VQA (ScienceQA, IconQA). It remains unclear whether FedNano generalizes to other multimodal tasks such as captioning, retrieval, grounding, or dialog.

**Questions:**

see weaknesses.

---

### Meta-Review · Area_Chair_Zara · 2026-01-06

**Summary:**

The reviewers' concerns are the novelty, significance and evaluations of this paper.

**Reviewer Concerns:**

they are still outstanding given no rebuttal is submitted.

**Reviewer Scores:**

none.

---

### Decision · Program_Chairs · 2026-01-26

Reject